

# Collective modes in superconductors including Coulomb repulsion

Joshua Althüser[⋆] and Götz S. Uhrig[†]

Condensed Matter Theory, TU Dortmund University,
Otto-Hahn Straße 4, 44227 Dortmund, Germany

⋆ joshua.althueser@tu-dortmund.de , † goetz.uhrig@tu-dortmund.de

## Abstract

We numerically study the collective excitations present in isotropic superconductors including a screened Coulomb interaction. By varying the screening strength, we analyze its impact on the system. We use a formulation of the effective phonon-mediated interaction between electrons that depends on the energy transfer between particles, rather than being a constant in a small energy shell around the Fermi edge. This justifies considering also rather large attractive interactions. We compute the system's Green's functions using the iterated equations of motion (iEoM) approach, which ultimately enables a quantitative analysis of collective excitations. For weak couplings, we identify the well-known amplitude (Higgs) mode at the two-particle continuum's lower edge and the phase (Anderson-Bogoliubov) mode at $\omega = 0$ for a neutral system, which shifts to higher energies as the Coulomb interaction is switched on. As the phononic coupling is increased, the Higgs mode separates from the continuum, and additional phase and amplitude modes appear, persisting even in the presence Coulomb interactions.

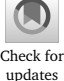

## 1  Introduction

Ever since its discovery over a century ago, superconductivity has captivated researchers due to its unique properties. The experimental observations defy any classical expectations by demonstrating perfect conductivity and diamagnetism. Especially the former effect offers a myriad of possible practical applications, spurring this ever-expanding field of research.

Our focus is on enhancing our understanding of collective excitations in superconducting systems. While the general effect of the Coulomb interaction is well understood, we seek to address the following key questions in this paper. How do collective excitations respond to screening effects? What relevant effects exist within the BCS channel? How does employing a more elaborate attractive interaction than standard BCS theory alter outcomes? How do stronger attractions affect collective excitations?

The first successful theoretical descriptions utilized an isotropic interaction that is attractive near the Fermi edge but vanishes elsewhere [1, 2]. Within this framework, two types of collective excitations emerge. One corresponds to amplitude fluctuations of the order parameter (Higgs mode) while the other to its phase fluctuations (Anderson-Bogoliubov mode). The former is located at the lower edge of the two-particle continuum. It has been observed both experimentally and theoretically by a plethora of previous studies [3–17]. Its excitation energy is widely interpreted as the minimum energy necessary to break up a Cooper pair.

In the absence of long-range Coulomb interactions, i.e., in a neutral superfluid, the phase mode becomes a gapless Goldstone mode. The inclusion of the Coulomb interaction, however, couples the phase of the order parameter to the electromagnetic field, shifting it toward the plasma frequency according to the common lore [2, 3, 5, 14, 17–24].

In this article, we numerically investigate superconducting systems. Omitting the Coulomb interaction and assuming a constant attractive interaction around the Fermi edge, we confirm the expected results exactly. We extend this analysis by incorporating an energy-dependent interaction and subsequently including Coulomb effects.

The attraction between two electrons in a conventional superconductor is typically attributed to an electron-phonon interaction that can be reinterpreted as an effective electron-electron interaction. Fröhlich presented the first description of this mechanism. While his result explains attraction at small energy transfers, it becomes repulsive at larger ones and exhibits linear divergences [25]. Subsequent *ansätze* provided alternative descriptions with more favorable characteristics, in that they are for no parameter regime repulsive and are confined to a small energy region [26–31]. While all these distinct formulations differ in their description of processes that have a finite energy transfer, they are identical for all real, i.e., on-shell, processes. Naturally, this has to be the case because these processes can be measured, at least in principle [30].

We adopt a result from a flow-equation approach, namely, a continuous unitary transformation (CUT) [27, 28, 31–34]. In the BCS channel, this formulation yields an attractive interaction proportional to a Heaviside function that depends on the magnitude of the energy transfer [31]. Thus, the interaction is attractive for small energy-transfers and vanishes otherwise. In this regard, it is better suited for the description of superconducting system's than Fröhlich's result as it encompasses neither repulsive regimes nor divergences in the BCS channel.

To compute the momentum-dependent superconducting order parameter we use a mean-field approximation to decouple the interaction terms. This yields gaps qualitatively and quantitatively similar to those predicted by standard BCS theory [1, 2], yet lacks the sharp cutoff observed at a certain distance from the Fermi edge. For small interaction strengths, computing the collective excitations using this interaction yields qualitatively the same results as a constant interaction. However, stronger interaction causes unexpected modes beyond the standard amplitude and phase mode to emerge from the two-particle continuum.

The Eliashberg formalism represents an alternative approach to superconductivity. Here, the retardation effects are included explicitly [35–38]. These are commonly used to argue why the phononic attraction dominates over the instantaneous Coulomb repulsion. In that regard, the basis transformation which we rely on in the derivation of the attractive interaction provides another advantage over Fröhlich's expression. Namely, it vanishes if the energy transfer is larger than the phonon energy $\omega_D$. Consequently, only scattering processes on a timescale larger than $1/\omega_D$ contribute. This is precisely what is meant by retardation. Hence, the employed formalism captures the essential effects of retardation [30].

Additionally, we incorporate the Coulomb interaction to study its impact on both the order parameter and the collective excitations. At the mean-field level, the Coulomb interaction effectively renormalizes the attractive interaction by a pseudopotential $\mu^*$ [2, 39] and causes the gap function to extend to arbitrary momenta. The order parameter also switches its sign close to the Fermi edge to avoid the repulsive nature of the Coulomb interaction as long as possible [39–41]. Investigating the collective excitations, we see the primary phase mode shifting towards high energies in accordance with expectations. This behavior can be continuously tracked by varying the screening strength of the Coulomb interaction. The aforementioned additional modes persist even after switching on the Coulomb repulsion.

Studying collective excitations lies beyond a simple mean-field approach. Therefore, we turn to the iterated equations of motion (iEoM) approach. Conceptually, this method begins with the Heisenberg equations of motion and selects a suitable operator basis, which is extended to represent all terms arising from commutation with the Hamiltonian. Naturally, this leads to an infinite hierarchy of equations that must be truncated for practical computations.

This procedure has been successfully applied in prior studies including comparisons with density matrix formalism for the quantum Rabi model [42] and applications to interaction quenches as well as collective excitations in Hubbard models [17, 43–45]. In this article, we will follow the procedure outlined in Ref. [17] to compute the Green's functions of the system and its collective excitations. Lifetime effects of the quasiparticles are neglected by this procedure. Nevertheless, due to the energy gap in the superconducting phase, lifetime effects are suppressed at low energies.

The remainder of the article is organized as follows: We briefly introduce the model under study in Sec. 2. In Sec. 3, we show our mean-field calculations and discuss the results. The study of the collective excitations is presented in Sec. 4. Finally, we summarize the results, conclude, and provide an outlook in Sec. 5.

## 2 Model

We focus on a rather general Hamiltonian at zero temperature given by

$$H = H_{\text{kin}} + H_{\text{Ph}} + H_{\text{C}} + H_{\text{BG}}, \tag{1}$$

where $H_{\text{kin}}$ describes the kinetic part, $H_{\text{Ph}}$ describes an effective electron-electron interaction mediated by phonons, $H_{\text{C}}$ describes the repulsive Coulomb interaction between electrons, and $H_{\text{BG}}$ describes the electronic interaction between the electrons and the atomic nuclei, which

we represent as a uniform positive background charge $\rho$. This term exactly cancels with the divergent Hartree contribution in $H_{\mathrm{C}}$ [46, 47]. In the remainder of this article, we will refer to contributions due to $H_{\mathrm{Ph}}$ as *phononic*. The individual terms are given by

$$H_{\mathrm{Kin}} = \sum_{k\sigma} \varepsilon_0(k) c_{k,\sigma}^\dagger c_{k,\sigma} \,, \tag{2a}$$

$$H_{\mathrm{Ph}} = \frac{1}{\mathcal{V}} \sum_{kk'\sigma} g(k,k') c_{k,\sigma}^\dagger c_{-k,-\sigma}^\dagger c_{-k',-\sigma} c_{k',\sigma} \,, \tag{2b}$$

$$H_{\mathrm{C}} = \frac{1}{2\mathcal{V}} \sum_{\substack{kk'q \\ \sigma\sigma'}} V(|q|) c_{k,\sigma}^\dagger c_{k',\sigma'}^\dagger c_{k'-q,\sigma'} c_{k+q,\sigma} \,, \tag{2c}$$

$$H_{\mathrm{BG}} = -\frac{1}{2} \sum_{k\sigma} V(0) \rho \, c_{k,\sigma}^\dagger c_{k,\sigma} \,. \tag{2d}$$

Here, $\mathcal{V}$ is the system's volume, $\varepsilon_0(k) = \hbar^2 k^2/(2m_e) - E_{\mathrm{F}}$ the free particle dispersion, $E_{\mathrm{F}}$ the Fermi energy, $\hbar$ the reduced Planck's constant, and $m_e$ the electron mass. The operator $c_{k,\sigma}^{(\dagger)}$ annihilates (creates) an electron with momentum $k$ and spin $\sigma$. The Coulomb potential is given by

$$V(q) = \frac{e^2}{\epsilon_0} \frac{1}{q^2 + k_s^2} \,, \tag{3}$$

where $e$ is the elementary charge, $\epsilon_0$ the vacuum permittivity, and $k_s$ the Thomas-Fermi screening wavevector. This is a standard procedure to capture the electronic screening effects in metals that result from diagrams as depicted in Fig. 1 [46, 48]. Moreover, this choice of potential provides a mathematical regularization to avoid divisions by zero. Often, the Coulomb interaction is considered to be reduced by a logarithm, cf. the theories presented by Tolmachev, and Morel and Anderson [39, 49]. As we present later, our method reproduces these results on the mean-field level for typical values of $k_s$. The case of unscreened Coulomb interaction is retrieved in the limit $k_s \to 0$ wherever this limit can be taken.

For comparison, we will also consider the Coulomb interaction only in the BCS channel

$$H_{\mathrm{C}}^{(\mathrm{BCS})} = \frac{1}{2\mathcal{V}} \sum_{kk'\sigma} V(|k-k'|) c_{k,\sigma}^\dagger c_{-k,-\sigma}^\dagger c_{-k',-\sigma} c_{k',\sigma} \,. \tag{4}$$

In this case, the background term is omitted as well because the Hartree contribution of the above term vanishes.

Obtaining an effective electron-electron interaction based on the physical electron-phonon interaction has been a major subject of past studies. Many *ansätze* have been proposed that serve this purpose. Moreover, the resulting effective interactions are not necessarily identical. For motivation, let $S$ be the generator of the unitary transformation that decouples the phononic and electronic subsystems in $\mathcal{O}(M^2)$. Then, one can add any interaction term in $\mathcal{O}(M^2)$ that conserves the number of phonons to $S$ while still achieving decoupling of the subsystems [30]. One of the best-known formulations was presented by Fröhlich [25]

$$g^{\mathrm{Fröhlich}}(k,k') = \frac{|M_{k'-k}|^2 \omega_{k'-k}}{(\varepsilon(k') - \varepsilon(k))^2 - \omega_{k'-k}^2} \,, \tag{5}$$

where $M_q$ is the electron-phonon coupling strength, $\omega_q$ is the phonon frequency, and $\varepsilon(k)$ represents the single-particle energy. This form has two significant disadvantages compared to other formulations, namely, that it is only attractive in a small energy interval and that it is divergent at $|\omega_{k'-k}| = |\varepsilon(k') - \varepsilon(k)|$.

Alternative *ansätze* resulted in expressions that remedied these issues by employing alternative basis transformations, see, e.g., Refs. [26–31]. The most important commonality among all these results is that they coincide for real, i.e., energy-conserving, processes. This agreement is essential since these processes are the ones that are observable experimentally, at least in principle [30]. In contrast, the virtual processes, i.e., the processes which do not conserve the total energy change depending on the specific basis transformation, may be altered by different unitary transformations. This freedom of choice is successfully exploited in continuous unitary transformations which we employ here to derive the effective interaction in the BCS channel [31]

$$g(\boldsymbol{k}, \boldsymbol{k}') = -\frac{|M_{\boldsymbol{k}'-\boldsymbol{k}}|^2}{\omega_{\boldsymbol{k}'-\boldsymbol{k}}} \Theta(\omega_{\boldsymbol{k}'-\boldsymbol{k}} - |\varepsilon(\boldsymbol{k}') - \varepsilon(\boldsymbol{k})|), \tag{6}$$

where $\Theta(k)$ is the Heaviside function and $\varepsilon(k) = \varepsilon_0(k) + \varepsilon_{\text{Fock}}(k)$ is the single-particle dispersion. The Fock energy $\varepsilon_{\text{Fock}}(k)$ arises from the mean-field treatment of $H_{\text{C}}$, see (15) below. We will, furthermore, restrict our discussion to a single phonon mode at the Debye frequency, i.e., $\omega_{\boldsymbol{k}'-\boldsymbol{k}} = \omega_{\text{D}}$, and a constant interaction strength $G := 2|M_{\boldsymbol{k}'-\boldsymbol{k}}|^2/\omega_{\text{D}} > 0$. This yields the interaction term

$$g_{\text{CUT}}(k, k') = -\frac{G}{2}\Theta(\omega_{\text{D}} - |\varepsilon(k) - \varepsilon(k')|). \tag{7}$$

Note that the factor 2 cancels with the spin summation in (2b). We will refer to this interaction as the *CUT interaction* throughout this article.

Within the framework of this article, we will only consider this interaction in the BCS channel. The full interaction exhibits divergent resonances at $|\omega_{\boldsymbol{k}'-\boldsymbol{k}}| = |\epsilon_{k'} - \epsilon_k|$ in the Fock channel. These resonances cannot be avoided by other types of transformations because they appear in the renormalization of the single-particle dispersion which is an on-shell process. Such resonances would likely be supressed by lifetime effects or smeared out when considering dispersive phonons. But this is beyond the scope of this article. Thus, we restrict our discussion to the interaction (2b) as described above.

We fix the Fermi wavevector to $\hbar k_{\text{F}}/\sqrt{m_e} = 4.25\sqrt{\text{eV}}$ throughout this article. This results in a Fermi energy of 8.5 eV to 9 eV depending on the screening wavevector $k_s$. Additionally, we fix the Debye frequency to $\omega_{\text{D}} = 10$ meV. These values are taken to be close to the actual parameters found in lead (Pb) [50]. We stress that our results are qualitatively independent of these parameters.

Furthermore, we define the dimensionless parameter

$$g := \rho_{\text{F}} G = G \frac{m_e k_{\text{F}}}{2\pi^2 \hbar^2}, \tag{8}$$

where $\rho_{\text{F}}$ is the density of states of the non-interacting system at the Fermi edge. The derivation of the effective interaction is perturbative in the second order of the electron-phonon coupling $M$. This perturbative approach is well justified if the matrix element $M$ is sufficiently small relative to the phonon energy $\omega_{\text{D}}$. We emphasize nevertheless, that large values of $g$ are possible if the density of states at the Fermi edge $\rho_F$ is large. Hence, it is justified to focus here on moderate values of $g$ of the order 1 to 10.

In the same manner, we define a dimensionless scaling ratio $\lambda$ for the screening wavevector

$$k_s := \lambda \sqrt{\frac{e^2 m_e}{3\pi^2 \hbar^2 \epsilon_0} k_{\text{F}}}. \tag{9}$$

The square-root expression is an estimate for the Thomas-Fermi wavevector in real materials [46]. Mathematically, this screening translates to the real-space potential

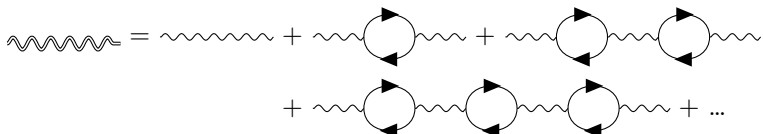

Figure 1: Diagramatic representation of the effective screened Coulomb interaction.

$V(r) \propto \exp(-k_s r)/r$. Physically, its existence can be motivated in the following manner: An electric field naturally causes a displacement of the charge carriers implying a polarization which in turn modifies the effective electric field. Diagrammatically, this process results in an effective interaction as depicted in Fig. 1, which corresponds to the aforementioned screened Coulomb potential [46].

Lastly, we define the following operators for brevity

$$f_{\boldsymbol{k}} := c_{-\boldsymbol{k},\downarrow} c_{\boldsymbol{k},\uparrow}, \qquad n_{\boldsymbol{k},\sigma} := c_{\boldsymbol{k},\sigma}^{\dagger} c_{\boldsymbol{k},\sigma}. \tag{10}$$

## 3 The superconducting gap function

### 3.1 General considerations

We use a mean-field approximation to decouple the interaction terms in (1). This will grant us access to the expectation values and the gap functions of interest.

On the mean-field level, (2b) boils down to a single non-vanishing term

$$\tilde{H}_{\mathrm{Ph}} = \sum_{\boldsymbol{k}} \Delta_{\mathrm{Ph}}(k) f_{\boldsymbol{k}}^{\dagger} + \mathrm{H.c.}, \tag{11}$$

where the phononic contribution to the superconducting gap function is given by

$$\Delta_{\mathrm{Ph}}(k) = \frac{1}{\mathcal{V}} \sum_{\boldsymbol{k}'} g(k,k') \langle f_{\boldsymbol{k}'} \rangle = \frac{1}{2\pi^2} \int_0^{\infty} \mathrm{d}k' k'^2 g(k,k') \langle f_{\boldsymbol{k}'} \rangle. \tag{12}$$

Repeating the same decoupling for the Coulomb repulsion $H_{\mathrm{C}}$ yields

$$\tilde{H}_{\mathrm{C}} = \sum_{\boldsymbol{k}} \Delta_{\mathrm{C}}(k) f_{\boldsymbol{k}}^{\dagger} + \mathrm{H.c.} + \sum_{\boldsymbol{k}\sigma} (\varepsilon_{\mathrm{Fock}}(k) + \varepsilon_{\mathrm{C}}(k)) n_{\boldsymbol{k},\sigma}. \tag{13}$$

Its additional contribution to the gap function reads

$$\Delta_{\mathrm{C}}(k) = \frac{e^2}{8\pi^2 \epsilon_0 k} \int_0^{\infty} \mathrm{d}q \langle f_q \rangle q \ln\left(\frac{k_s^2 + (q+k)^2}{k_s^2 + (q-k)^2}\right). \tag{14}$$

Note that this term has the opposite sign of the term in (12).

Assuming $\langle n_{k,\sigma} \rangle = \Theta(k_{\mathrm{F}} - |\boldsymbol{k}|)$ at zero temperature, the Fock energy reads

$$\varepsilon_{\mathrm{Fock}}(k) = -\frac{e^2}{4\pi^2 \epsilon_0} k_{\mathrm{F}} \bigg[ 1 + \frac{k_s}{k_{\mathrm{F}}} \left( \arctan\left(\frac{k-k_{\mathrm{F}}}{k_s}\right) - \arctan\left(\frac{k+k_{\mathrm{F}}}{k_s}\right) \right)$$
$$+ \frac{k_{\mathrm{F}}^2 - k^2 + k_s^2}{2kk_{\mathrm{F}}} \ln\left(\frac{k_s^2 + (k_{\mathrm{F}}+k)^2}{k_s^2 + (k_{\mathrm{F}}-k)^2}\right) \bigg]. \tag{15}$$

In the superconducting phase, however, $\langle n_{k,\sigma} \rangle$ is more complicated than assumed above. We define the difference between these expression as $\delta_n(k) := \Theta(k_{\mathrm{F}} - k) - \langle n_{k,\sigma} \rangle$. Notably, this

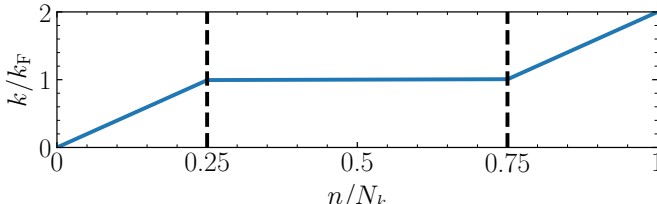

Figure 2: Schematic representation of the numerical discretization showing the magnitude of $k$ versus the discretization index $n$. The black dashed lines show the cutoff points at which the discretization switches from coarse to fine and vice versa. Note that there is still a minute slope for $n \in [N_k/4, 3N_k/4]$.

quantity is mostly close to 0 and has a finite contribution only close to the Fermi edge. Nevertheless, the fact that $\langle n_{k,\sigma} \rangle$ in the superconducting phase is not a Heaviside function implies a small correction to the Fock energy given by

$$\varepsilon_{\mathrm{C}}(k) = \frac{e^2}{8\pi^2 \epsilon_0 k} \int_0^\infty \mathrm{d}q \delta_n(q) q \ln\left(\frac{k_s^2 + (q+k)^2}{k_s^2 + (q-k)^2}\right), \qquad (16)$$

which is typically of the order of a few µeV.

Lastly, note that the Fock energy vanishes if we restrict our calculations to the BCS channel (4). At the mean-field level, the vanishing of the Fock energy is also the main difference compared to utilizing the entire Coulomb interaction. The most notable effect of the vanishing of the Fock energy is that the Fermi energy shifts by $\varepsilon_{\mathrm{Fock}}(k_{\mathrm{F}}) \approx -0.5\,\mathrm{eV}$ for a weak screening of $\lambda = 10^{-4}$. Naturally, this affects $\rho_{\mathrm{F}}$ and, therefore, the magnitude of the gap function. Nevertheless, its shape remains qualitatively the same. Accordingly, to avoid repetition, we will limit the discussion of the mean-field results to the full Coulomb interaction (2c).

For the numerics, we notice that all of the above quantities merely depend on the absolute value $k = |\mathbf{k}|$ due to the rotational symmetry of the model. Therefore, we discretize $k$ using $N_k$ individual points. Due to the long-range nature of the Coulomb interaction, we need to extend our discretization to a large interval, theoretically even from 0 to $\infty$. But in practice, the choice $k \in [0, 2k_{\mathrm{F}}]$ yields good results.

The main physics can be observed around the Fermi edge so that an equidistant discretization in the entire interval would be inefficient. It is more efficient to use a fine mesh around the Fermi edge that utilizes $N_k/2$ individual points and is cut off at $k_{\mathrm{F}} \pm x_{\mathrm{cut}} \omega_{\mathrm{D}}/k_{\mathrm{F}}$. Beyond this cutoff, the discretization is coarser with $N_k/4$ additional discretization points in each direction. A schematic representation is shown in Fig. 2.

The choice of $x_{\mathrm{cut}}$ is arbitrary to some extent. If the gap is small, a small value is beneficial to resolve the relevant part of the gap function in more detail. If the gap is large, however, we require a larger $x_{\mathrm{cut}}$ so that the entire relevant part is included. The relevant metric is the order parameter's value at its maximum $\Delta_{\mathrm{max}}$. In practice, we choose $x_{\mathrm{cut}} = 10$ for $\Delta_{\mathrm{max}} < 4\,\mathrm{meV}$, $x_{\mathrm{cut}} = 20$ for $\Delta_{\mathrm{max}} < 14\,\mathrm{meV}$, and $x_{\mathrm{cut}} = 25$ beyond that. These cutoffs allow for some leeway, i.e., varying these values slightly does not impact the results, as long as one chooses a large enough $N_k$. In practice, $N_k = 30000$ produces good results for $x_{\mathrm{cut}} = 25$, while $N_k = 20000$ is sufficiently precise for $x_{\mathrm{cut}} = 10$ and 20.

## 3.2   Discussion of the numerical results

Let us begin by omitting the Coulomb interaction, i.e., setting $e = 0$. In this case, the gap function only includes the phononic contribution. Often, the effective electron-electron interaction is assumed to be constant in a small region around the Fermi edge [1, 2, 51]. It is then

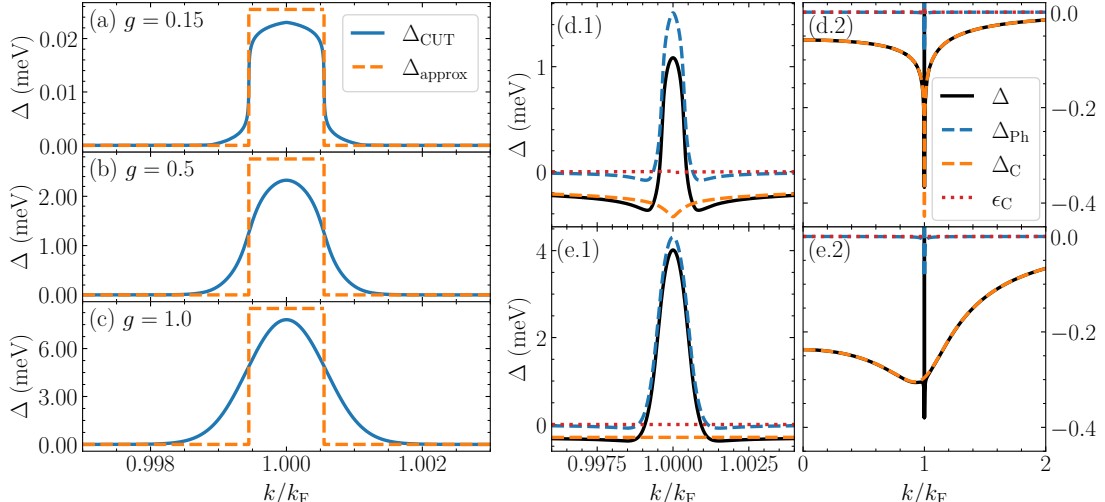

Figure 3: (a-c) The superconducting gap function without Coulomb interaction. The blue solid line shows the result for the CUT interaction (7) while the orange dashed line depicts the data for the BCS interaction (17). In each case, we set $N_k = 8000$ and $g$ according to the text in each panel. (d-e) The superconducting gap function including the Coulomb interaction. The blue and orange dashed lines depict the phononic (12) and Coulomb (14) contributions to the gap function, which itself is represented by the black solid line. The red dotted line shows the correction to the Fock energy (16). We used $\lambda = 10^{-4}$ for panels (d) and $\lambda = 1$ for panels (e). Panels (1) show the gap functions in a small region around the Fermi edge, while panels (2) depict them across the entire numerical range. Note the difference in energy scales between the panels. For both cases, we set $N_k = 20000$ and $g = 0.8$.

approximated by

$$g_{\text{BCS}}(k, k') := -\frac{G}{2}\Theta(\omega_D - |\varepsilon(k)|)\Theta(\omega_D - |\varepsilon(k')|). \tag{17}$$

We will refer to this interaction as the *BCS interaction* throughout this article. We briefly study this case as well to compare the results to those obtained by using the proper description (7). Three exemplary gap functions for Eq. (17) and the CUT interaction (7), are depicted in Fig. 3 (a-c). We set $N_k = 8000$ and $g = \{0.15, 0.5, 1.0\}$. The gap functions themselves are qualitatively generic. Increasing $g$ increases their magnitude and increasing $\omega_D$ increases their width.

As expected, both gap functions are restricted to a small region around the Fermi edge. However, the gap function corresponding to the CUT interaction is slightly wider and does not have hard boundaries. It further exhibits a smaller magnitude. The gap function's general form approaches the form obtained via the BCS interaction in the limit $g \to 0$.

Often, the focus of studies lies on the gap's magnitude at the Fermi edge. To this end, the approximation (17) is certainly sufficient for a qualitative description of the physics, although the results differ slightly. Nevertheless, we will see important differences between both approaches later during the discussion of the collective excitations.

Next, we add the Coulomb interaction which results in considerable changes to the gap function. A plot of two exemplary gap functions is shown in Fig. 3 (d) and (e). Here, we used $N_k = 20000$ and $g = 0.8$. In the top panel (d), the screening is set to $\lambda = 10^{-4}$ while it is set to $\lambda = 1$ in the bottom panel (e). Again, the functions are qualitatively generic with regard to the dependence on $g$ and $\omega_D$. Naturally, increasing the screening also increases the gap's magnitude, see Fig. 3 (d.1) and (e.1). Also, as mentioned before, the correction to the Fock energy $\varepsilon_C$ is negligibly small.

A striking feature that arises due to the Coulomb interaction is that the gap function stretches over all $k$ values. This occurs for any screening strength. While its main contribution is still confined to a small region around the Fermi wavevector, it approaches a constant value for $k \to 0$ and follows a $1/k^2$ behavior as $k \to \infty$. This statement is proven in Appendix A.

Furthermore, due to the repulsive nature of the Coulomb interaction, the gap function has a radial node when said interaction starts dominating over the phononic one. Similar behavior is found within the framework of the Eliashberg theory and the Anderson-Morel model [37, 40, 41, 52].

At this point, it should be mentioned that the complex phase of the gap function can still be fixed to be real. We repeated the calculation allowing for an arbitrary complex phase, but the results are essentially identical. Instead of changing the sign at the nodal points, the complex phase of the gap function jumps by $\pi$.

Considering a small screening, i.e., $\lambda = 10^{-4}$ in panels (d) of Fig. 3, results in a rather narrow valley in $\Delta_\mathrm{C}$ around the Fermi edge. It is noteworthy, that this result is essentially identical to the one obtained by omitting the screening entirely because no screening is necessary to deal with the logarithmic singularities as the latter are smoothed by the integration. While this part of our calculations can still be easily performed without screening, the later parts require at least a small screening to avoid numerically dividing by 0.

The aforementioned valley in $\Delta_\mathrm{C}$ disappears entirely if we use the screening wavevector introduced in Ref. [46], i.e., $\lambda = 1$ in panel (e.1). In this case, the contribution of $\Delta_\mathrm{C}$ around the Fermi edge is essentially a constant. This corroborates the qualitative statements of the commonly used result by Morel and Anderson [39] that a screened Coulomb interaction essentially boils down to a constant pseudopotential $\mu$ around the Fermi edge, modifying the relevant interaction strength $g \to g - \mu$ [14, 27, 41, 48, 53]. Bogoliubov provided a first expression for the pseudopotential [2]. We multiply it by the density of states at the Fermi edge $\rho_\mathrm{F}$ to obtain the dimensionless expression

$$\mu = \frac{e^2 \rho_F}{4 \epsilon_0 k_\mathrm{F}^2} \ln\left( \frac{k_s^2 + 4k_\mathrm{F}^2}{k_s^2} \right). \tag{18}$$

Later, Morel and Anderson showed that, in the weak-coupling limit $\mu \ll 1$, the relevant interaction strength is actually given by $g - \mu^*$ with

$$\mu^* = \frac{\mu}{1 + \mu \ln\left( \frac{E_\mathrm{F}}{\omega_\mathrm{D}} \right)}, \tag{19}$$

which they finally used to obtain [39]

$$\ln\left( \frac{\Delta_\mathrm{max}}{2\omega_\mathrm{D}} \right) = -\frac{1}{g - \mu^*}. \tag{20}$$

Using this expression, we can fit our results for $\Delta_\mathrm{max}$. Note, however, that Eq. (20) was derived under the assumption of a constant BCS interaction. We already saw in Fig. 3 (a-c) that the CUT interaction (7) leads to a slightly reduced gap value at the Fermi edge. Thus, we introduce two additional fit parameters $\alpha$ and $\beta$ and fit to

$$\ln\left( \frac{\Delta_\mathrm{max}}{2\omega_\mathrm{D}} \right) = \ln\alpha - \frac{\beta}{g - \mu^*}. \tag{21}$$

Here, $\alpha$ represents a constant ratio relating $\Delta_\mathrm{max}$ to $\omega_\mathrm{D}$, while $\beta$ serves as the scaling factor in the exponential. Based on (20), we expect these values to be at least close to 1. In Fig. 4, we show the fits of the data excluding and including the Coulomb interaction. Without the Coulomb interaction, we investigated both the BCS interaction (17) and the CUT interaction (7). Including the Coulomb repulsion, we show only the CUT interaction with the screening being set to $\lambda = 1$ and $\lambda = 10^{-4}$. The corresponding fit parameters are provided in Table 1.

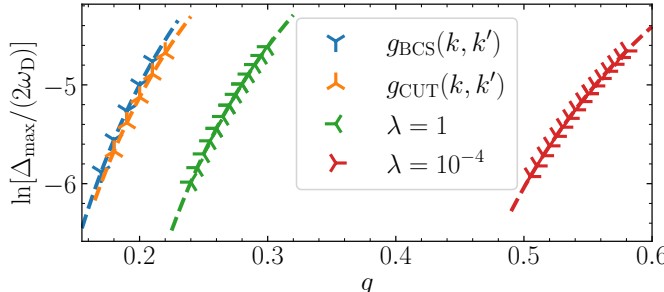

Figure 4: Logarithm of $\Delta_{\mathrm{max}}$ in units of $2\omega_{\mathrm{D}}$ versus the phononic coupling strength $g$. The markers represent the evaluated data points while the lines show the best fit in accordance with Eq. (21). The data represented by the blue and orange markers excludes the Coulomb interaction and used the BCS interaction (17) and the CUT interaction (7), respectively. For the data in green and red, we used the CUT interaction and included the Coulomb interaction while fixing the screening to $\lambda = 1$ and $\lambda = 10^{-4}$. The fit parameters are given in Table 1.

Table 1: Fit parameters for the fits shown in Fig. 4. The data sets are obtained by (i) using the BCS interaction (17), (ii) using the CUT interaction (7), (iii) additionally including the Coulomb repulsion with $\lambda = 1$, and (iv) using $\lambda = 10^{-4}$. We also show the values of the pseudopotential proposed by Bogoliubov (18), denoted as $\mu$, and the pseudopotential $\mu^*$ of Morel and Anderson (19).

| Data | Fit parameters | | | Predictions | |
|------|----------------|---|---|-------------|---|
| | $\alpha$ | $\beta$ | $\mu^*_{\mathrm{Fit}}$ | $\mu$ | $\mu^*$ |
| (i) | $1.0000 \pm 0.0003$ | $0.9999 \pm 0.0001$ | $(8 \pm 9) \cdot 10^{-6}$ | $0$ | $0$ |
| (ii) | $0.7804 \pm 0.0009$ | $0.9679 \pm 0.0004$ | $0.00154 \pm 0.00003$ | $0$ | $0$ |
| (iii) | $0.707 \pm 0.012$ | $1.056 \pm 0.006$ | $0.0526 \pm 0.0005$ | $0.0688$ | $0.0469$ |
| (iv) | $0.31 \pm 0.04$ | $1.02 \pm 0.06$ | $0.291 \pm 0.006$ | $0.3428$ | $0.1035$ |

Since the approximation (17) is just the standard BCS formulation, we expect the BCS behavior $\ln(\Delta_{\mathrm{max}}/(2\omega_{\mathrm{D}})) = -1/g$, which is within numerical accuracy exactly what we find. The CUT interaction (7) deviates from that only slightly, namely in $\alpha$, i.e., the ratio between $\Delta_{\mathrm{max}}$ and $\omega_{\mathrm{D}}$ which is marginally smaller now. Notably, the exponential scaling factor $\beta$ agrees well with the expectation $\beta = 1$ in all cases.

Switching on the Coulomb interaction provides more insight. Firstly, using a typical screening $\lambda = 1$ results in a $\mu^*$ that is in reasonable agreement with Eq. (19). A small screening of $\lambda = 10^{-4}$, however, yields considerable deviations. Nevertheless, this is no contradiction to the original statement as it was derived for $\mu \ll 1$, which is no longer given here.

If we assume a constant interaction strength and neglect the Coulomb interaction, we obtain a constant order parameter $\Delta(k)$ around the Fermi edge. Then, it is trivial to see that the quasiparticle dispersion

$$E(k) = \sqrt{(\varepsilon(k) + \varepsilon_{\mathrm{C}}(k))^2 + \Delta^2(k)}, \tag{22}$$

has its global minimum at the $k_{\mathrm{F}}$. However, this is not necessarily the case if the order parameter varies. In principle, it could diminish faster than the single-particle dispersion $\varepsilon(k)$ rises for $k \neq k_{\mathrm{F}}$. We do observe this behavior for large enough $g$, see Fig. 5. In the upper panels (a-c), we show the quasiparticle energy (22) relative to $\Delta_{\mathrm{max}}$ for various $g$. The Coulomb interaction

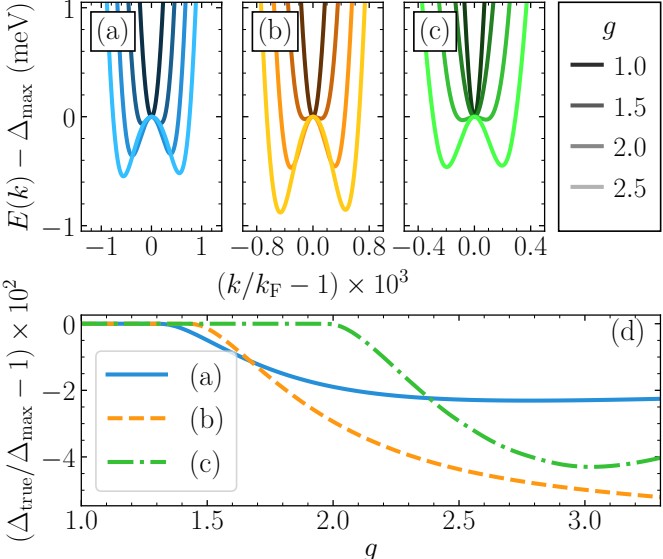

Figure 5: The upper row (a-c) depicts the quasiparticle dispersion (22) relative to the order parameter's peak value $\Delta_{\max}$. The individual panels have (a) no Coulomb interaction, (b) the screening set to $\lambda = 1$, and (c) $\lambda = 10^{-4}$. Brighter lines represent stronger phononic interactions in accordance with the legend in the top right. Notably, the energy minimum is not located at $k_F$ for large $g$ but shifts to $k < k_F$. The plot is not entirely symmetrical about $k_F$ due to the parabolic dispersion, though the effect is minuscule. The lower panel (d) depicts the system's true energy gap $\Delta_{\text{true}}$ in units of order parameter's peak value $\Delta_{\max}$ versus the phononic interaction strength $g$. The individual lines represent the form of the Coulomb interaction as described above in accordance with the legend. Note that the $x$-axis begins at $g = 1$ because $\Delta_{\text{true}} = \Delta_{\max}$ holds for moderate $g$.

is excluded in panel (a) and included with a screening of $\lambda = 1$ in panel (b) and $\lambda = 10^{-4}$ in panel (c). For all cases, there exists a $g$ with $\min(E(k)) \neq k_F$, though $|k_{\min} - k_F|/k_F \ll 1$ holds. We define the energy at this minimum as $\Delta_{\text{true}}$, and compare it to $\Delta_{\max}$ in panel (d) of Fig. 5. Initially, the ratio is a constant at $\Delta_{\text{true}}/\Delta_{\max} - 1 = 0$ since both quantities are identical for moderate $g$. Subsequently, the minimum shifts away from $k_F$ and $\Delta_{\text{true}}$ diminishes. Nevertheless, this behavior slows down and even reverses as $g$ is increased further.

Lastly, Fig. 6 shows the order parameter's peak value $\Delta_{\max}$ depending on the screening $\lambda$. Naturally, it grows as the screening is increased, but it does not seem to follow any elementary function. Expectedly, the function approaches a constant value for both cases $\lambda \to 0$ and $\lambda \to \infty$. The general behavior does not vary qualitatively for various values of $g$.

## 4 Collective excitations

### 4.1 General considerations

Analyzing collective excitations is more involved and lies beyond the simple mean-field approach discussed in the previous section. We follow the iEoM approach as discussed in Ref. [17] to obtain the relevant Green's functions.

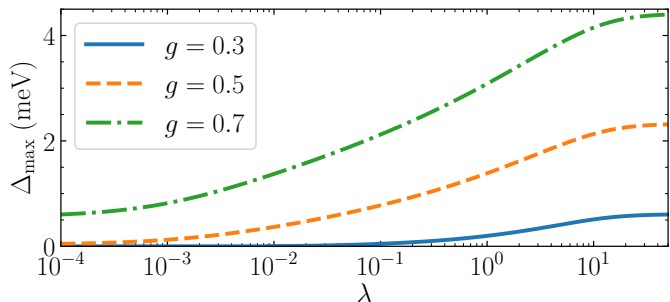

Figure 6: Plot of the order parameter's peak value $\Delta_{\max}$ as a function of the screening $\lambda$. The screening is scaled logarithmically. The phononic interaction strength is set to three distinct values, see the legend. Otherwise, the parameters are identical to the ones in Fig. 3 (d) and (e).

To investigate Higgs and phase modes, we compute the Green's function with respect to the operators

$$\mathfrak{A}_{\mathrm{Higgs}} := \frac{1}{\sqrt{\mathcal{V}}} \sum_{\boldsymbol{k}} \left( f_{\boldsymbol{k}}^{\dagger} + f_{\boldsymbol{k}} \right), \tag{23a}$$

$$\mathfrak{A}_{\mathrm{Phase}} := \frac{\mathrm{i}}{\sqrt{\mathcal{V}}} \sum_{\boldsymbol{k}} \left( f_{\boldsymbol{k}}^{\dagger} - f_{\boldsymbol{k}} \right). \tag{23b}$$

Note that both operators are fully isotropic, i.e., any modes excited by them are modes without angular momentum. Furthermore, these operators have no center-of-mass momentum. Consequently, the corresponding excited collective excitations are excitations at zero momentum. To compute the corresponding Green's functions, we choose an operator basis $\mathfrak{B}$ for the iEoM that contains the operators

$$\mathcal{A}_k := \frac{1}{\sqrt{\mathcal{V}}} \sum_{\boldsymbol{q}} \delta(k - |\boldsymbol{q}|) \left( f_{\boldsymbol{q}}^{\dagger} + f_{\boldsymbol{q}} \right), \tag{24a}$$

$$\mathcal{P}_k := \frac{1}{\sqrt{\mathcal{V}}} \sum_{\boldsymbol{q}} \delta(k - |\boldsymbol{q}|) \left( f_{\boldsymbol{q}}^{\dagger} - f_{\boldsymbol{q}} \right), \tag{24b}$$

$$\mathcal{N}_k := \frac{1}{\sqrt{\mathcal{V}}} \sum_{\boldsymbol{q}} \delta(k - |\boldsymbol{q}|) \left( n_{\boldsymbol{q},\uparrow} + n_{-\boldsymbol{q},\downarrow} \right), \tag{24c}$$

for all $k$ under consideration. Note that (23a) and (23b) are easily represented by our choice of basis. For example, consider

$$\mathfrak{A}_{\mathrm{Phase}} = \frac{\mathrm{i}}{\sqrt{\mathcal{V}}} \int \mathrm{d}k \sum_{\boldsymbol{q}} \delta(k - |\boldsymbol{q}|) \left( f_{\boldsymbol{q}}^{\dagger} - f_{\boldsymbol{q}} \right) = \mathrm{i} \int \mathrm{d}k \mathcal{P}_k. \tag{25}$$

The next step is to compute the dynamical matrix $\mathcal{M}$ and norm matrix $\mathcal{N}$

$$\mathcal{M}_{ij} = (O_i | [H, O_j]), \tag{26a}$$
$$\mathcal{N}_{ij} = (O_i | O_j), \tag{26b}$$

with each $O_i \in \mathfrak{B}$ and the pseudo-scalar product $(A|B) := \langle [A^{\dagger}, B] \rangle$. We evaluate the occurring commutators with respect to the full Hamiltonian (1). Then, we take the expectation values with respect to the mean-field Hamiltonian and make use of Wick's theorem to compute quartic expectation values.

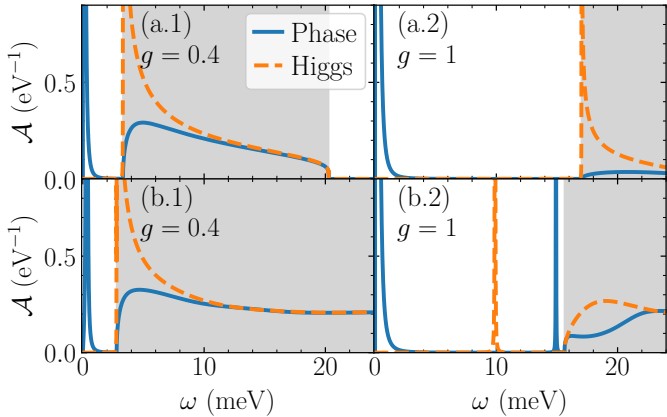

Figure 7: Spectral functions $\mathcal{A}_{\text{Phase}}(\omega)$ (blue solid line) and $\mathcal{A}_{\text{Higgs}}(\omega)$ (orange dashed line) without the Coulomb interaction. A small imaginary shift is introduced to the energy $\omega \rightarrow \omega + i\delta$ to plot the $\delta$-peaks below the continuum as Lorentzians with $\delta = 10^{-3}$ meV. In the upper panels (a), we used the BCS interaction (17) while we used the CUT interaction (7) in the lower ones. The grey area marks the two-particle continuum. Prominently, we can identify the phase mode at $\omega = 0$, and, for small $g$ the Higgs mode at $\omega = 2\Delta_{\text{max}}$. The Higgs mode moves below the gap and additional modes appear as $g$ is increased, but only for the CUT interaction (7).

These matrices can be analytically related to the Fourier-transformed Green's functions

$$\mathcal{G}_\alpha(z = \omega + i0^+) = -\frac{i}{\hbar}\int_0^\infty \mathrm{d}t\, e^{izt}\langle[\mathfrak{A}_\alpha(t), \mathfrak{A}_\alpha^\dagger(0)]\rangle\,, \qquad \alpha \in \{\text{Higgs}, \text{Phase}\}\,. \tag{27}$$

Numerically, we perform a Lanczos tridiagonalization and obtain a continued fraction expansion of Eq. (27) that we terminate using the square-root terminator [17, 54, 55]. Then, the spectral functions are obtained by $\mathcal{A}_\alpha = -(1/\pi)\mathfrak{I}[\mathcal{G}_\alpha(\omega + i0^+)]$.

As the relevant physics occurs in the vicinity of the Fermi edge, we restrict the analysis of the collective modes to this region. Specifically, the $k$-integrals in this section are restricted to the fine mesh around the Fermi edge. This procedure allows us to obtain precise data for the collective modes that have an energy similar to or smaller than the gap function. However, it carries the caveat that the two-particle continuum is only resolved up to energies in the order of $x_{\text{cut}}\omega_D$. One can expect that the same applies to other features at high energies, i.e., that it will not be possible to provide precise numerical data for collective excitations at high energies. Still, we will see for example that the phase mode shifts to large energies as we include the Coulomb repulsion.

Lastly, it should be mentioned, that we require at least some screening for these calculations because the commutators yield otherwise divergent terms. As an example consider $\langle[f_{k'}^\dagger - f_{k'}, [H, f_k - f_k^\dagger]]\rangle$. One of the resulting terms is $V(|k - k'|)\langle f_k\rangle\langle f_{k'}\rangle$, which has a singularity at $k = k'$. Because these terms are contributions to individual matrix elements of matrices that we need to manipulate in a non-trivial manner, we cannot lift nor easily avoid these kinds of singularities and ultimately require a finite screening. But we find that the results are not very sensitive to the exact value of $\lambda \ll 1$.

## 4.2  Classification of collective excitations

As before, we start the discussion by omitting the Coulomb interaction, i.e., $e = 0$. First, we compute the spectral functions using the BCS interaction (17). The results are shown in the

upper panels of Fig. 7. Here, we shift the energy into the complex plane by a small constant $\omega \to \omega + i\delta$ to resolve the peaks below the two-particle continuum. In this article, we set $\delta = 10^{-3}$ meV. The interaction strength was set to $g = 0.4$ in the left panels and to $g = 1$ in the right panels, which yields a gap of $\Delta_{\max} \approx 1.65$ meV and $\Delta_{\max} \approx 8.51$ meV, respectively. The spectral functions each exhibit a single peak, $\mathcal{A}_{\text{Higgs}}(\omega)$ at $\omega = 2\Delta_{\max}$ and $\mathcal{A}_{\text{Phase}}(\omega)$ at $\omega = 0$. These features are well-known in the literature and are commonly associated to the Higgs and the phase mode, respectively [2–24].

To analyze peaks below the continuum, we fit $\Re[\mathcal{G}_{\alpha}(\omega)]$ in close vicinity to them. For the phase peak at $\omega = 0$, we find $\Re[\mathcal{G}_{\text{Phase}}(\omega)] \propto 1/\omega^2$, which translates to $\mathcal{A}_{\text{Phase}}(\omega) \propto \delta'(\omega)$ due to the Kramers-Kronig relations. This can be interpreted as two peaks at $\pm\omega_0$ with $\omega_0 \to 0$ because the spectral function with respect to a Hermitian bosonic operator is antisymmetric [17].

By fitting $\mathcal{A}_{\text{Higgs}}(\omega)$ close to the peak, we find that it falls off like $1/\sqrt{\omega - 2\Delta_{\max}}$. These results are consistent with previous findings. For instance, the order parameter behaves as $\Delta(t) \propto \cos(2\Delta t)/\sqrt{t}$ shortly after a weak interaction quench [3,4,7,16] which is the Fourier transform of our result. Also note that there is no qualitative difference between the results obtained for different values of $g$.

Next, we examine the CUT interaction (7). The results are displayed in the lower panels of Fig. 7. For small to moderate interaction strengths $g$, the results do not differ significantly from those obtained using the BCS interaction. The Higgs mode continues to behave asymptotically like an inverse square root while the phase mode behaves like $\delta'(\omega)$. Notably, the most significant difference is that $\Delta_{\max}$ is slightly reduced. Specifically, the gaps are now $\Delta_{\max} \approx 1.39$ meV and $\Delta_{\max} \approx 7.81$ meV instead of $\Delta_{\max} \approx 1.65$ meV and $\Delta_{\max} \approx 8.51$ meV, respectively.

For larger values of $g$, however, the Higgs mode shifts to energies below $2\Delta_{\max}$. Increasing the interaction strength further spawns even more modes below the continuum in both $\mathcal{A}_{\text{Higgs}}(\omega)$ and $\mathcal{A}_{\text{Phase}}(\omega)$, see Fig. 7 panel (b.2). Note that this only happens if we use the CUT interaction (7) and not if we use the BCS interaction (17). This indicates that there are intricate mechanisms at play that are not captured by the simple approximation (17).

To distinguish between the modes, we refer to the Higgs mode and the phase mode that can be associated with those found in literature as *primary* modes. The additional modes that emerge below the continuum are referred to as *secondary* modes.

As anticipated, the primary phase mode remains located at $\omega = 0$, since there are still no long-range Coulomb interactions present [56, 57]. Furthermore, we fit the real part of the Green's functions near the peaks below the continuum. Doing so yields the same behavior as before for the primary phase peak and $\Re[\mathcal{G}_{\text{Phase}}(\omega)] \propto 1/(\omega - \omega_0)$ for the peaks at finite energies $\omega_0 < 2\Delta_{\max}$ indicating $\mathcal{A}_{\alpha}(\omega) \propto \delta(\omega - \omega_0)$. In the same manner, it is straightforward to determine the spectral weights of these excitations.

To observe the evolution of the collective excitations upon varying $g$, we plot the spectral functions $\mathcal{A}(\omega)$ in Fig. 8. We build on the previous analysis and represent the occurring $\delta$-peaks by Gaussian bell curves and the $\delta'$-peaks by the derivative of a Gaussian bell. The energy $\omega$ is plotted on the $y$-axis and the phononic coupling strength on the $x$-axis. The coloring indicates the magnitude of the spectral functions. The modes appear as sharp bright lines, while the two-particle continuum is faintly visible due to the comparatively low magnitude of the spectral functions inside that region.

To guide the eye, we also plot the lower edge of the continuum at $2\Delta_{\text{true}}$ as a cyan dotted line. Notably, the primary phase mode remains at $\omega = 0$ for all $g$ as expected.

The primary Higgs mode first evolves along the lower edge of the continuum until it smoothly breaks away from it for larger $g$. This phenomenon has been observed previously by Barankov and Levitov in the context of order parameter oscillations after an interaction quench [58]. Note that the non-trivial momentum dependence of the interaction is essential

**SciPost** SciPost Phys. 19, 067 (2025)

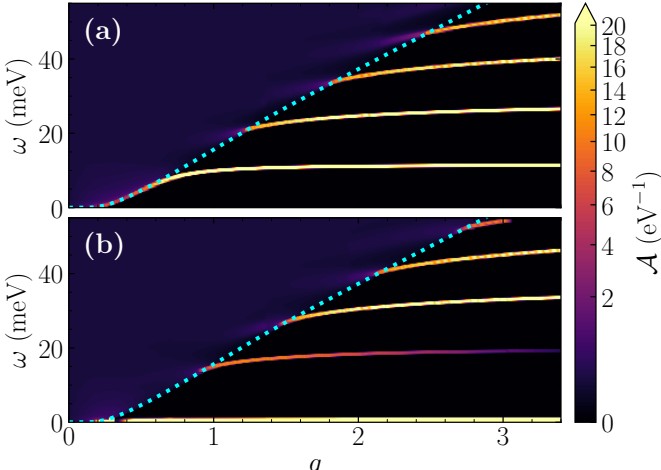

Figure 8: Spectral functions (a) $\mathcal{A}_{\mathrm{Higgs}}(\omega)$ and (b) $\mathcal{A}_{\mathrm{Phase}}(\omega)$ without the Coulomb interaction, i.e., $e = 0$. The magnitude is represented by the color scale. The peaks below the continuum are mathematically $\delta$-peaks and are represented by Gaussian bell curves with the same weight and width $\sigma = 0.05\,\mathrm{meV}$. The only exception to this is the peak at $\omega = 0$ in $\mathcal{A}_{\mathrm{Phase}}(\omega)$. It is described by the derivative of a $\delta$-peak and therefore represented by the derivative of a Gaussian bell curve. The coupling strength $g$ is indicated on the $x$-axis. The cyan dotted line shows the lower edge of the two-particle continuum at $2\Delta_{\mathrm{true}}$. The evolution of the modes is clearly visible as they emerge smoothly from the two-particle continuum. Note that the color scale is non-linear to better depict the occurring modes.

to observing this detachment of the Higgs mode. For the constant BCS interaction, the Higgs mode remains at $2\Delta_{\mathrm{max}}$. After that, the mode has a minute upward tendency in energy. At the same time, additional modes emerge smoothly from the continuum. These secondary modes appear in an alternating fashion first in the phase and then in the Higgs spectral function. Their energy gain remains small afterward. We will further investigate these modes in Sec. 4.4.

## 4.3 Effects of the Coulomb interaction

To establish a baseline for the following discussions, we include the full Coulomb interaction with a nearly vanishing screening $\lambda = 10^{-4}$. The resulting spectral functions are depicted in Fig. 9. Setting $g = 1$, results in $\Delta_{\mathrm{max}} \approx 2.2\,\mathrm{meV}$. The Higgs mode is diminished in magnitude, but its asymptotic behavior remains $\mathcal{A}_{\mathrm{Higgs}}(\omega) \propto 1/\sqrt{\omega - 2\Delta_{\mathrm{max}}}$. The phase mode, however, is absent from the low-energy regime and can now be found at high energies, as one would expect.

Increasing $g$ spawns the same secondary modes as before, but this effect only manifests at significantly larger values of $g$ compared to before. This shift can be attributed to the Coulomb repulsion, which reduces the gap. Consequently, a stronger attraction $g$ is required to achieve the same $\Delta_{\mathrm{max}}$ as before. Therefore we postulate that the governing mechanism behind the occurrence of these excitations is not the attraction strength $g$ itself, but the magnitude of the gap. We will delve into this statement later, cf. Sec. 4.4.

Note that there is still no mode at $\omega = 0$ in Fig. 9 panel (b). Additionally, the lowest-lying Higgs mode appears at lower energies than the lowest-lying phase mode. With this in mind, we want to investigate how the phase mode shifts to higher energies. (i) Is the BCS channel of the interaction sufficient to explain this behavior? (ii) Does it evolve smoothly in terms of the screening?

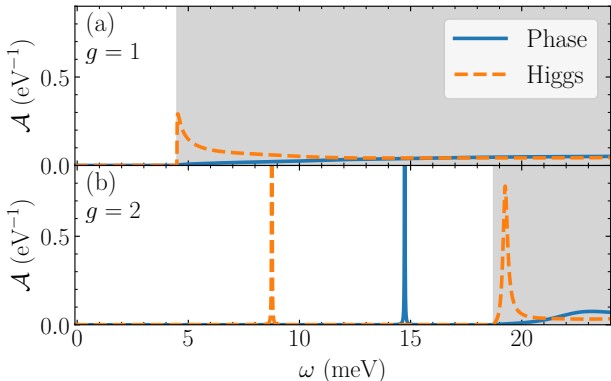

Figure 9: Spectral functions $\mathcal{A}_{\mathrm{Phase}}(\omega)$ (blue solid line) and $\mathcal{A}_{\mathrm{Higgs}}(\omega)$ (orange dashed line) with the Coulomb interaction. A small imaginary shift is introduced to the energy $\omega \to \omega + i\delta$ to resolve the peaks below the continuum; $\delta = 10^{-3}\,\mathrm{meV}$. Panel (a) and (b) depict the results for $g = 1$ and $g = 2$, respectively. For both cases, we used an almost vanishing screening $\lambda = 10^{-4}$. The Higgs mode persists after the introduction of the Coulomb interaction. The phase mode at $\omega = 0$, however, disappears as expected and shifts to high energies beyond the regime depicted in this plot. Increasing $g$ again spawns secondary modes, similar to what is presented in Fig. 7.

To address question (i), we use the Coulomb interaction only in the BCS channel (4) and perform the computations for various $g$. We show the resulting data in Fig. 10 (a) and (b). As before, the upper panels (a) depict $\mathcal{A}_{\mathrm{Higgs}}(\omega)$ and the lower panels (b) depict $\mathcal{A}_{\mathrm{Phase}}(\omega)$. In the left column (1), we used a screening of $\lambda = 1$. For the right one (2), we set $\lambda = 10^{-4}$.

The results for both cases are qualitatively similar. The secondary modes emerge from the continuum essentially as they did without any Coulomb interactions. The primary Higgs mode, however, exhibits an altered behavior. It picks up a downward tendency for both screenings and even reaches $\omega = 0$. For $\lambda = 10^{-4}$, this point is $g \approx 2.54$, which is marked by a magenta dashed line, see Fig. 10 (a.2). This behavior indicates instabilities of the system. Simultaneously, the dynamical matrix $\mathcal{M}$ acquires a negative eigenvalue for $g > 2.54$. However, this matrix must be positive semidefinite if the system is in thermal equilibrium [17]. Thus, we conclude that beyond that point the system favors a different phase, which is beyond the scope of this article, though.

Turning to $\mathcal{A}_{\mathrm{Phase}}(\omega)$ in the lower panels, we notice that the primary phase mode is present below the continuum. For both screenings, it shifts to higher energies linearly with $g$. At a certain point, it crosses the next-higher phase mode, which exhibits a downward tendency just as the Higgs mode did. Such level crossings do not occur typically, as couplings between the two levels introduce level repulsion. Combined with the fact that the primary phase mode remains at relatively small energies, we conclude that the BCS channel for the Coulomb interaction is insufficient to fully describe the relevant physics. Nevertheless, including the Coulomb interaction only in the BCS channel suffices to lift the primary phase mode to a finite energy.

Hence, we turn to the study of the full Coulomb interaction (2c) in Fig. 10 (c) and (d). First considering $\lambda = 1$ in the left panels (1), we find that the modes are mostly similar to the ones presented earlier. However, the level crossing is now absent and has been replaced by an anticrossing.

The energies of the lowest Higgs and the lowest phase mode still have a downward tendency at larger $g$ for both screening strengths. Beyond the region shown in the plot for $g > 3.65$, the same behavior as before occurs, i.e., the primary Higgs mode shifts to $\omega = 0$ and the system becomes unstable. We will discuss this in more detail later, cf. Sec. 4.4 and Fig. 12 (a.3).

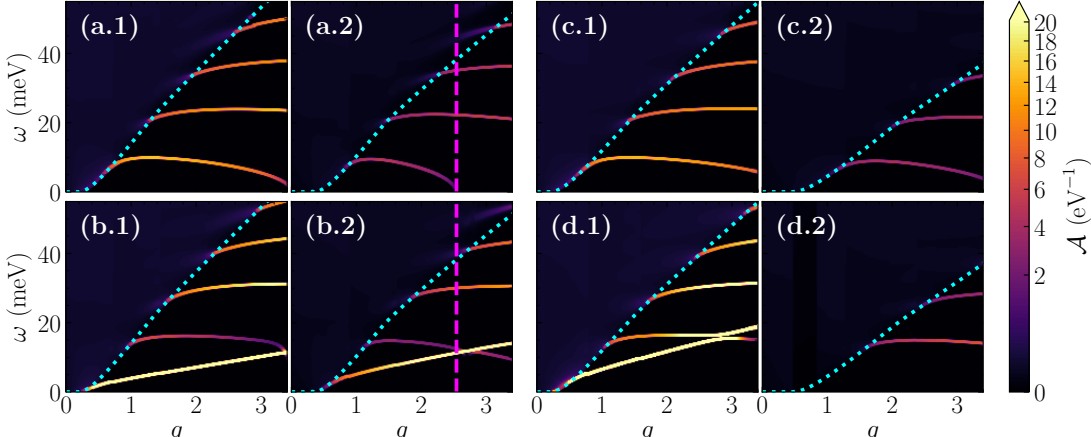

Figure 10: Same as Fig. 8, except that the Coulomb interaction is included. We used only in the BCS channel (4) for panels (a) and (b), while we used the full Coulomb interaction (2c) for panels (c) and (d). The upper panels (a) and (c) depict $\mathcal{A}_{\mathrm{Higgs}}(\omega)$ and the lower panels (b) and (d) depict $\mathcal{A}_{\mathrm{Phase}}(\omega)$. Panels (1) show the data for a screening of $\lambda = 1$ and panels (2) the data for $\lambda = 10^{-4}$. Notably, the phase mode is no longer located at $\omega = 0$ but at finite energy below the two-particle continuum for all cases except panel (d.2). Additionally, when the lowest-lying Higgs mode in panel (a.2) reaches 0 (magenta dashed line at $g \approx 2.54$), the dynamical matrix $\mathcal{M}$ picks up a negative eigenvalue, hinting at an instability of the system.

A central point is that we confirm that the primary phase mode vanishes for all $g$ from the low-energy regime. Thereby, we are able to provide a numerical calculation that is fully in unison with the Anderson-Higgs mechanism [3, 18, 21]. Moreover, the BCS channel merely couples the Cooper pairs to one another while the full interaction accounts for the interaction between all electrons. Combining this with the fact that the BCS channel is insufficient to describe the Anderson-Higgs mechanism, we can confirm that the phase mode is coupled to the collective motion of the electrons by the inclusion of the Coulomb interaction. This coupling occurs specifically due to the inclusion of a nearly unscreened interaction with proper long-range behavior.

The next goal is to understand how the panels (c.1) and (d.1) of Fig. 10 evolve into the panels (c.2) and (d.2) upon decreasing $\lambda$, thereby answering question (ii). To this end, we fix the phononic interaction strength to $g = 0.5$ and vary the screening $\lambda$. The results are shown in Fig. 11 (a) and (b) where the $\lambda$-axis is scaled logarithmically. As before, the upper panel (a) depicts $\mathcal{A}_{\mathrm{Higgs}}(\omega)$, and the lower one (b) depicts $\mathcal{A}_{\mathrm{Phase}}(\omega)$. Of course, the gap is affected by the screening since a stronger screening implies weaker repulsion and therefore a larger gap. Other than that, there is little impact on the Higgs mode. It remains located at $\omega = 2\Delta_{\mathrm{max}}$ and it only gains in magnitude as the gap itself grows. Note that the Higgs mode does not detach from the two-particle continuum because the parameters do not result in large gaps. In contrast, tracing the phase mode from vanishing screening toward strong screening, we observe its smooth evolution down from high energies. It leaves the continuum around $\lambda = 1$, though the precise value depends on the specific choice of system parameters.

To better understand this process, we show the phase mode's position $\omega_{\mathrm{P}}$ in units of $2\Delta_{\mathrm{max}}$ versus the screening in a double-logarithmic plot in Fig. 11 (c) and (d). We depict the result for (c) $g = 0.5$ and (d) $g = 0.7$. The plot is restricted to the screenings for which $\omega_{\mathrm{P}}$ is inside the continuum because the fits only work in this range. Within the continuum, $\omega_{\mathrm{P}}$ behaves almost like $1/\lambda$, possibly with logarithmic corrections, as indicated by the fits represented by orange dashed lines.

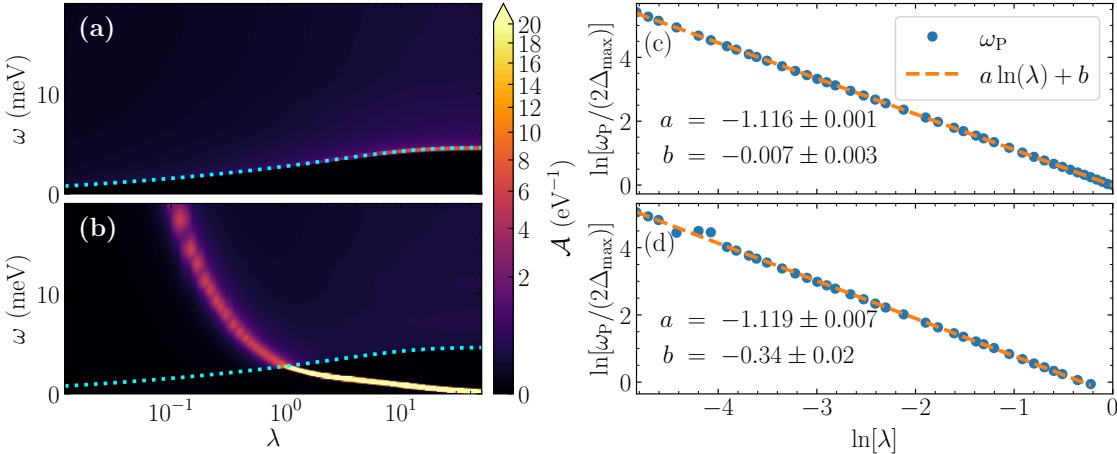

Figure 11: The left panels depict the spectral functions (a) $\mathcal{A}_{\text{Higgs}}(\omega)$ and (b) $\mathcal{A}_{\text{Phase}}(\omega)$ with the full Coulomb interaction at different screenings $\lambda$. The phononic coupling strength is fixed to $g = 0.5$ and $\lambda$ is varied logarithmically on the $x$-axis. The magnitude is represented by the color scale. The color scale is non-linear to depict the occurring modes better. The peaks below the continuum are mathematically $\delta$-peaks and are represented by Gaussian bell curves with the same weight and width $\sigma = 0.05$ meV. The cyan dotted line shows the lower edge of the two-particle continuum at $2\Delta_{\text{true}} = 2\Delta_{\text{max}}$. The phase mode smoothly shifts down from high energies as the screening is increased. The Higgs mode is located at $\omega = 2\Delta_{\text{max}}$ for all screenings because the parameters do not yield large gaps. The right panels show double logarithmic plots of the position $\omega_{\text{P}}$ of the resonance in $\mathcal{A}_{\text{Phase}}(\omega)$ in units of $2\Delta_{\text{max}}$ depending on the screening $\lambda$. The phononic interaction strength is set to (c) $g = 0.5$ and (d) $g = 0.7$. The blue circles mark the data used for the fits which are represented by the orange dashed lines. The plot depicts a range of $\lambda$ so that $\omega_{\text{P}}$ is inside the continuum. Here, $\omega_{\text{P}}$ follows an almost straight line with a slope independent of $g$.

Contrary to expectations, this behavior does not stop at the plasma frequency, which in our case is a few electronvolts. Rather, $\omega_{\text{P}}$ continues to increase in the same manner as described above as $\lambda$ decreases. We believe that this is due to numerical limitations. We cannot resolve the high-energy regime properly as our approach, including the choice of the numerical mesh and cutoff, is focused on the low-energy regime. Therefore, the mode likely behaves as depicted in Fig. 11 for moderate screenings but will deviate in the limit $\lambda \to 0$ in accordance with literature predictions [3, 18, 21].

Lastly, it bears mentioning that tuning $\omega_{\text{D}}$ has no significant effect on the results. We present the data for this statement in Appendix B.

## 4.4 Emergence of secondary modes

The last question we seek to address is how the various modes behave as they emerge from the two-particle continuum. These secondary modes emerge both with and without Coulomb interactions but at different $g$. As formulated previously, the working hypothesis is that the modes emerge at certain threshold values of the gap $\Delta_{\text{max}}$, not at specific values of $g$. If the Coulomb repulsion is stronger, i.e., less screened, a larger value of $g$ is requried to induce the same gap $\Delta_{\text{max}}$.

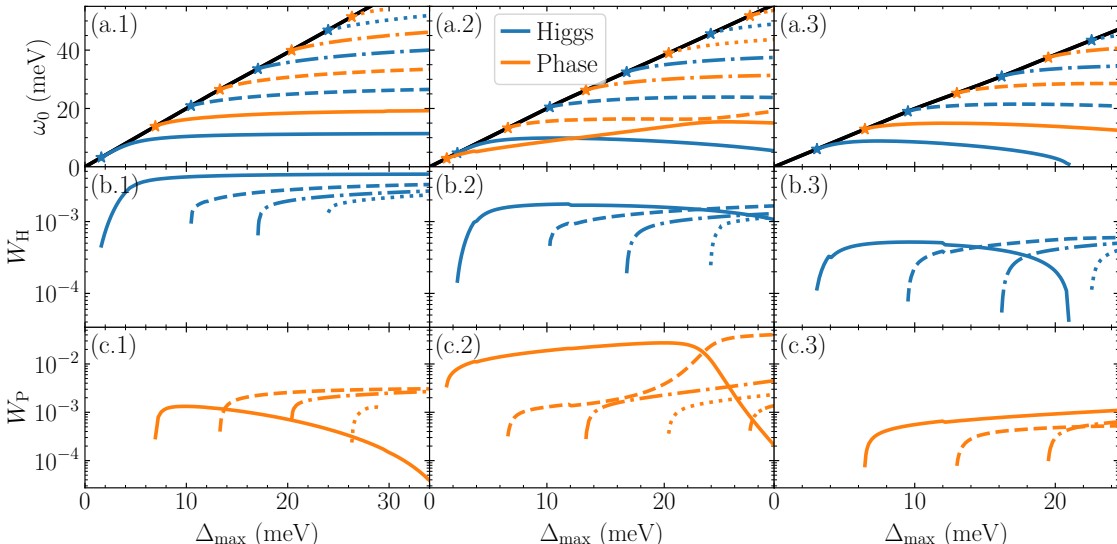

Figure 12: The panels in the first row (a) show the energies $\omega_0$ of the Higgs (blue) and phase (orange) modes depending on $\Delta_{max}$. The black line marks the lower edge of the two-particle continuum. In column (1) the Coulomb interaction is omitted while it is included in the other two columns; in (2) with $\lambda = 1$ and in (3) with $\lambda = 10^{-4}$. The $y$-axes of the last two rows are scaled logarithmically and depict the weight of the peaks in (b) $\mathcal{A}_{\text{Higgs}}(\omega)$ and (c) $\mathcal{A}_{\text{Phase}}(\omega)$. Lines with different styles correspond to one another within a column, e.g., the solid blue lines in (a.1) and in (b.1) depict the data for the same mode. For the left panels (1), we omit the primary phase mode at $\omega = 0$ due to its distinct behavior as discussed in the previous section. The peaks emerge with initially vanishing weights.

Figure 12 depicts the peak positions $\omega_0$ (panels (a)) and their weights $W$ (panels (b) for $\mathcal{A}_{\text{Higgs}}(\omega)$ and (c) for $\mathcal{A}_{\text{Phase}}(\omega)$) as a function of the maximum of the gap. The left panels (1) depict the data for no Coulomb interaction. The remaining panels show the data for the full Coulomb interaction with $\lambda = 1$ in the center panels (2) and $\lambda = 10^{-4}$ in the right panels (3). We choose different linestyles to distinguish between the various modes. The blue lines depict the modes in $\mathcal{A}_{\text{Higgs}}(\omega)$ and the orange lines those in $\mathcal{A}_{\text{Phase}}(\omega)$. Additionally, we omitted the primary phase mode at $\omega = 0$ from the left panels (1) because it has distinct properties from the secondary ones.

At the point of emergence, the peaks have vanishing weights; the mode's weight remains within the continuum. However, as the modes move away from the continuum, the weights of the peaks increase rapidly. Most of the excitations' weights essentially saturate beyond which they exhibit only a slight upward tendency.

One exception is the first phase mode in panel (c.1) as its weight tends towards 0. A similar behavior occurs in panel (c.2). Here, the first and the second phase mode experience a level repulsion for large $g$. At this point, their weights swap, i.e., the lower-lying mode loses most of its weight which then tends to 0. Concurrently, the other mode gains a significant weight and then behaves like the other modes, i.e., its weight has only a slight upward tendency.

The first Higgs mode tends towards $\omega = 0$ and loses its weight for both cases that include Coulomb interactions (panels 2 and 3). As mentioned in the previous section, this kind of behavior is a precursor to a phase transition. When dealing with superconducting systems, one commonly employs the Landau theory for continuous phase transitions. In this case, the free energy $F$ of the system is approximated by a polynomial in the order parameter, see Fig. 13.

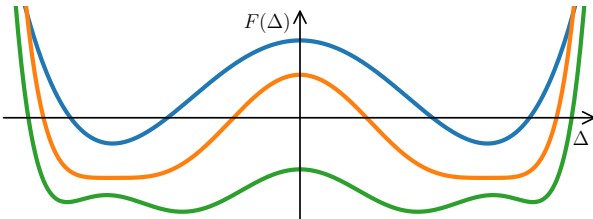

Figure 13: Schematic representation of the free energy $F$ as a function of the superconducting order parameter $\Delta$. The standard Landau theory results in the blue curve. The orange curve represents the situation when the primary amplitude mode becomes soft. Here, changing the order parameter barely affects the free energy. The green curve depicts a possible landscape of the free energy afterward when multiple minima might be present.

The standard Landau theory results in parabolic minima at the equilibrium value of $\Delta$ as sketched by the blue curve [59]. However, small perturbations to the magnitude of the order parameter would barely affect the free energy if the free-energy landscape becomes flat around the minima as depicted by the orange curve. This is likely what happens as the primary Higgs mode becomes soft and therefore small amplitude fluctuations require minimal energy to be excited. Beyond this point, the free energy might exhibit multiple minima, as sketched by the green curve.

Upon closer inspection of the upper panels (a) in Fig. 12, we anticipate that the various modes appear at about the same $\Delta_{\mathrm{max}}$, essentially independent of the Coulomb interaction. To confirm this, we depict specifically these values in Fig. 14. Here, the $\Delta_{\mathrm{max}}$ at which the modes emerge is plotted along the $y$-axis, and the modes are counted along the $x$-axis. The square markers represent Higgs modes while the circles represent phase modes. The colors represent the three cases of the Coulomb interaction as discussed above. The black markers are their averages. The error bars represent one standard deviation. We omitted the primary phase modes due to their special behavior. The obvious conclusion is that corresponding excitations emerge at almost the same gaps.

We fit the averages linearly so that the slopes $s_\alpha$ represent the spacing in $\Delta_{\mathrm{max}}$ between the emergence of the modes. This yields $s_{\mathrm{Higgs}} = (6.98 \pm 0.26)\,\mathrm{meV}$ and $s_{\mathrm{Phase}} = (6.68 \pm 0.15)\,\mathrm{meV}$, respectively. By taking the average of these two values using inverse-variance weighting, we obtain $s = (6.76 \pm 0.13)\,\mathrm{meV}$. This quantity means that for every $s$ that the gap grows, we expect to find one additional mode per channel.

Notably, repeating these calculations for $\hbar k_{\mathrm{F}}/(2m) = 5\sqrt{\mathrm{eV}}$ yields almost the same plot. In this case, we obtain the slopes $s_{\mathrm{Higgs}} = (7.04 \pm 0.25)\,\mathrm{meV}$, $s_{\mathrm{Phase}} = (6.62 \pm 0.16)\,\mathrm{meV}$, and $s = (6.75 \pm 0.14)\,\mathrm{meV}$. This corroborates our hypothesis that the governing parameter behind the appearance of the secondary modes is the magnitude of the superconducting gap.

To our knowledge, these kinds of subgap excitations have not been comprehensively studied before. Similar excitations, however, can be found in $d$-wave superconductors. In this case, they can be attributed to an additional rotational degree of freedom due to the nodal structure of such systems [15, 60].

Contrary to this, our model does not break rotational symmetry. Nevertheless, we observe a non-trivial *radial* energy landscape for large phononic interaction strengths. We presume that the emerging secondary modes relate to this non-trivial radial dependence.

Specifically, the second Higgs mode emerges from the continuum around the same $g$ at which the quasiparticle dispersion forms minima at $k \neq k_{\mathrm{F}}$ as discussed in Sec. 3, cf. Fig. 5. Nevertheless, this particular kind of peculiarity can be observed only for the second Higgs mode.

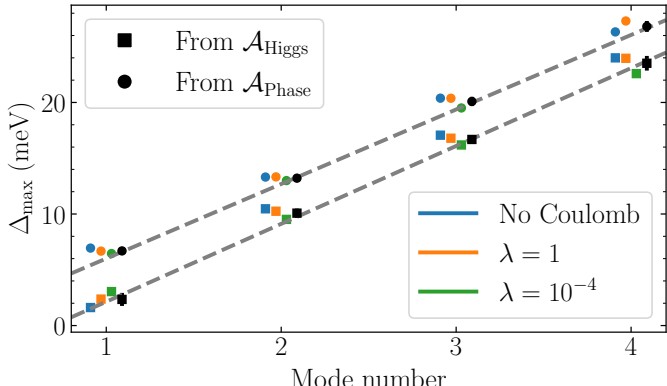

Figure 14: Peak value of the gap $\Delta_{max}$ at which the secondary modes emerge from the continuum. The squares represent Higgs modes, while the circles represent phase modes. The primary phase modes have been omitted from this plot as they have distinctive behavior. The $x$-axis denotes the mode number. The data points are shifted slightly from integer values to improve visibility. The colors indicate the type of Coulomb interaction as described by the legend. The black symbols are the averages of the corresponding three colored ones. Their error bars represent the standard deviation. The gray dashed lines are linear fits to the black markers. Their slopes $s_\alpha$ describe the spacing in $\Delta_{max}$ between the emergence of individual modes. The slopes are $s_{Higgs} = (6.98 \pm 0.26)\,\text{meV}$ and $s_{Phase} = (6.68 \pm 0.15)\,\text{meV}$.

## 5 Conclusion

The goal of this article was to investigate collective excitations in superconductors using a more complete description than the standard BCS theory. To this end, we expanded upon the standard constant attraction by employing an energy-transfer-dependent interaction (7) in the BCS channel derived via a continuous unitary transformation. This attractive interaction is called effective phononic interaction since it stems from the electron-phonon coupling. Additionally, we included the long-range Coulomb interaction which causes the phase mode to shift toward high energies, in accordance with the Anderson-Higgs mechanism.

We started by studying the static mean-field properties of the system. Omitting the Coulomb interaction, the superconducting gap function behaves qualitatively similar to the one obtained by the BCS theory. The gap is only finite in the close vicinity of the Fermi edge. Beyond this region, it tends to zero continuously. Furthermore, the gap exhibits a slightly diminished magnitude. Upon switching on Coulomb interactions, the gap function exhibits radial nodes close to the Fermi edge where it switches its sign as function of momentum. We observed good agreement with the result based on the pseudopotential $\mu^*$ introduced by Morel and Anderson [39] if the interaction is screened. Notable deviations occur for nearly unscreened interactions. Here, we obtained a significantly larger value for $\mu^*$. However, this is no contradiction because the derivation of Morel and Anderson is valid for the weak-coupling regime, which is not applicable in the case of small screenings.

Due to the non-trivial momentum dependence of the gap function, we found that the quasiparticle dispersion does not display its minimum at the Fermi edge for large interaction strengths. Instead, the minimum shifts to slightly lower momenta because the gap function falls off faster than the electron dispersion rises.

The heart of the present endeavor was the study of collective excitations enabled by computing Green's functions via the iterated equations of motion approach. By revisiting the standard BCS theory without including the Coulomb interaction, we successfully reproduced estab-

lished results from the literature. Specifically, we identified the Higgs mode at the lower edge of the two-particle continuum and confirmed the phase mode at zero energy, validating our computational framework. Next, we went beyond the standard BCS coupling by employing the energy-transfer-dependent interaction (7) also for studying the collective excitations.

Our findings align with the previous results for small and moderate coupling strengths. For larger coupling strengths, the Higgs mode detaches from the continuum. Such a detachment implies that the Higgs mode becomes dissipationless as has been discussed by Barankov and Levitov in the context of persisting oscillations of the order parameter after an interaction quench [58]. The momentum dependence of the interaction is crucial for the emergence of this phenomenon. Our results perfectly align with this as we did not see such a detachment when we considered the constant BCS interaction. Upon increasing the coupling strength further, additional modes emerge in both the amplitude and phase spectral functions. To our knowledge, there has not been a comprehensive study of these isotropic subgap excitations previously. Within the context of $d$-wave superconductors, similar excitations were found and attributed to the nodal structure of the gap function [15, 60]. In our case, we believe that the non-trivial radial energy landscape for large phononic interaction strengths is linked to the additional, secondary modes.

Similar additional modes appear if one considers non-isotropic excitations. In particular, Bardasis and Schrieffer used the conservation of angular momentum $L$ in isotropic systems and showed that collective excitations corresponding to different $L$ can occur below $2\Delta$ [61]. These so-called Bardasis-Schrieffer modes are typically observed if there are strong subdominant pairing channels in the system that correspond to a different pairing symmetry such as $d$-wave superconductivity [62–65]. While the ideas are similar, our results are distinct because we compute the Green's functions with respect to isotropic operators. Therefore, the modes we identify are excitations with $L = 0$. In analogy to the standard hydrogen problem, our secondary modes likely correspond to different radial dependences analogous to different values of the principal quantum number for hydrogen.

To study the effects of the Coulomb interaction itself, we followed two approaches. First, we restricted the Coulomb interaction to the BCS channel (4). In this case, there is little difference between the results for various screenings beyond changes of the magnitude of the gap. This approach is sufficient to lift the phase mode from 0 to a finite energy, however it remains below the gap. Contrary to that, the Anderson-Higgs mechanism states that the phase mode should be located at high energies around the plasma frequency [3, 18, 21]. Furthermore, the primary phase mode crosses with the next-higher phase mode which is at odds with the expectations for level repulsion. Altogether, we conclude that the inclusion of the Coulomb interaction only in the BCS channel is insufficient to describe the relevant physics.

Therefore, we considered the full Coulomb interaction (2c) next. Crucially, we find that the primary phase mode shifts towards large energies continuously as the screening strength is decreased in line with theoretical predictions. Our method is desigend for the low-energy regime and does not yield precise numerical values at large energies towards the upper band edge, i.e., in the limit of vanishing screening. Nevertheless, it provides strong support for established theories of the Anderson-Higgs mechanism [3, 18, 21].

For moderately strong screenings, we observe that the phase mode remains below the gap. Importantly, the previously noted level crossing becomes an anticrossing as expected from level repulsion. Thus, we confirm that it is important to include the full Coulomb interaction in all channels because it couples the phase of the order parameter to electronic density fluctuations which are not captured by the BCS channel alone.

We emphasize that the additional secondary modes persist even for the full inclusion of the Coulomb interaction. But they appear for larger values of the coupling $g$. Their points of emergence appear universal when they are scanned as function of the maximal value of the gap

function $\Delta_{\mathrm{max}}$ instead of as function of the coupling $g$. Then, these secondary modes appear at regular intervals of $\Delta_{\mathrm{max}}$, independent of whether Coulomb interactions are included or not. This observation suggests that these modes represent a robust feature and will also occur in more elaborate treatments and in presence of further interactions.

Lastly, we observed the primary Higgs mode becoming soft which we interpret as the system becoming unstable at large interaction strengths. At present, we cannot conclude whether this feature is spurious or whether it indicates a quantum phase transition by condensation of the primary Higgs mode to novel phases.

The present study calls for further investigations. A particularly intriguing investigation in this respect is an experimental one: Are there systems with strong BCS coupling in which experimental signatures of the predicted secondary modes can be discerned? Terahertz spectroscopy appears to be an appropriate tool for this search.

On the theoretical side, at zero total momentum, the present study can be extended by allowing non-isotropic excitations in the isotropic system, i.e., with finite angular momentum which represent Bardasis-Schrieffer modes [61–65]. It is conceivable that such modes become soft even before the secondary isotropic phase and Higgs modes emerge or become soft.

Another extension suggesting itself is to consider excitations not only at zero momentum, but at finite momentum [14]. In this way, the dispersive behavior can be addressed. But it must be pointed out that this requires more comprehensive discretizations than at zero momentum where it was sufficient to discretize the magnitude of the momentum.

To corroborate our finding of secondary modes alternative approximations can be employed, for instance a diagrammatic approach to the electron-phonon system without unitary reduction to an electron-only model. This would amount up to analyzing the corresponding Eliashberg theory. Thus, a future study could compare the results obtained here with those from the Eliashberg theory [35–38] to understand the differences and similarities between the two approaches. Such a comparison could not only validate our findings, but also reveal if and how lifetime effects influence the excitation spectra.

Last but not least, one can pass from the fully isotropic model to lattice models with reduced point group symmetries of the Fermi surfaces. Then, the collective excitations above a ground state of different symmetry, such as $d$-wave symmetry [15], can be investigated.

## Acknowledgments

We acknowledge very helpful discussions with D. Hering, J. Stolze, I. Eremin, J. Schmalian, and M. Sigrist.

**Funding information** This research was partially funded by the MERCUR Kooperation in project KO-2021-0027.

## A   Limiting behavior of the gap function

In this section, we analytically show how the gap function behaves in the limits $k \to 0$ and $k \to \infty$. First, we note that the phononic contribution (7) vanishes in both limits. Thus, we

will only discuss the Coulomb contribution (14). For $k \to 0$, consider

$$\lim_{k \to 0} \Delta_{\mathrm{C}}(k) = \lim_{k \to 0} \frac{e^2}{8\pi^2 \epsilon_0 k} \int_0^\infty \mathrm{d}q \langle f_q^\dagger \rangle q \ln\left( \frac{k_s^2 + (q+k)^2}{k_s^2 + (q-k)^2} \right) \tag{A.1a}$$

$$= \frac{e^2}{8\pi^2 \epsilon_0} \lim_{k \to 0} \frac{1}{k} \int_0^\infty \mathrm{d}q \langle f_q^\dagger \rangle q \left( \frac{4kq}{k_s^2 + q^2} + \mathcal{O}(k^3) \right) \tag{A.1b}$$

$$= \frac{e^2}{2\pi^2 \epsilon_0} \int_0^\infty \mathrm{d}q \langle f_q^\dagger \rangle \frac{q^2}{k_s^2 + q^2} \tag{A.1c}$$

$$= \text{const.} \tag{A.1d}$$

Similarly, we obtain for $k \to \infty$

$$\Delta_{\mathrm{C}}(k \gg k_{\mathrm{F}}) = \frac{e^2}{8\pi^2 \epsilon_0 k} \int_0^\infty \mathrm{d}q \langle f_q^\dagger \rangle q \left( \frac{4q}{k} + \mathcal{O}\left( \frac{1}{k^3} \right) \right) \tag{A.2a}$$

$$= \frac{e^2}{2\pi^2 \epsilon_0} \frac{1}{k^2} \int_0^\infty \mathrm{d}q q^2 \langle f_q^\dagger \rangle \tag{A.2b}$$

$$\propto \frac{1}{k^2}. \tag{A.2c}$$

# B   Dependence on the Debye frequency

In this section, we briefly discuss how the Debye frequency $\omega_{\mathrm{D}}$ affects the collective excitation. As an example, we vary $\omega_{\mathrm{D}}$ in Fig. 15. For the left column (1), we omitted the Coulomb interaction and set $g = 0.4$, while for the right column (2), we included it with $\lambda = 10^{-4}$ and $g = 1$. The gap scales linearly with $\omega_{\mathrm{D}}$, as discussed before. Therefore, we plot the $\omega/(2\Delta_{\max})$ on the $y$-axis. The continuum begins then exactly at 1 for moderate gap sizes. All modes are at a constant position in this plot. We checked this for various other parameters and found that this behavior is generic.

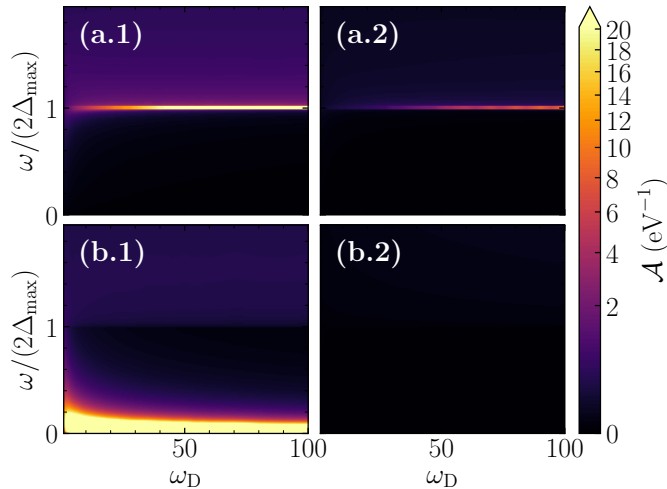

Figure 15: Spectral functions (a) $\mathcal{A}_{\mathrm{Higgs}}(\omega)$ and (b) $\mathcal{A}_{\mathrm{Phase}}(\omega)$ depending on $\omega_{\mathrm{D}}$. The $y$-axis is scaled by $\omega/\Delta_{\max}$ to show the modes' behavior more clearly. The left column shows the data if no Coulomb interaction is present, while the right one includes it. The phononic coupling strength is set to $g = 0.4$ and $g = 1$, respectively.

In conclusion, changing the Debye frequency has no significant effect on the results presented within this article.

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
