# Peer review of "Collective modes in superconductors including Coulomb repulsion"

_SciPost Physics, doi:SciPost Phys. 19, 067 (2025)_

## Round 1 · Referee Report · Anonymous (Referee 1) · 2025-5-28

Strengths

  1. It is a creative paper in its physically justified choice of the interaction with quite nontrivial result
  2. The results are correct - I was able to verify the shift of the Higgs mode myself using somewhat different method.
  3. The authors provide an important insight for the origin of the effect which is nicely illustrated in Fig. 5

Weaknesses

There is only one weakness here:

  1. The manuscript does not offer anything substantially new conceptually. In fact, the work uses essentially the same idea - momentum dependent interaction - as the one that has been put forward in a paper by Barankov and & Levitov [arXiv: 0704.1292] and thoroughtly forgotten since then. Specifically Barankov and Levitov have also found that momentum dependent interaction lead to the shift of the Higgs mode below the single-particle threshold. I believe the authors should cite and discuss the results of that work since, as I mentioned above, conceptually it has quite a bit of overlap with the present study although the methodology is a bit different.

Report

I think this manuscript contains enough important results to be published provided that the authors cite and discuss the work I mentioned above

Recommendation

Publish (easily meets expectations and criteria for this Journal; among top 50%)

  • validity: top
  • significance: high
  • originality: good
  • clarity: high
  • formatting: perfect
  • grammar: perfect

Author:  Joshua Althüser  on 2025-06-23  [id 5590]

(in reply to Report 1 on 2025-05-28)
Category:
answer to question
reply to objection

We thank the Referee for their time and kind and positive report and for making us aware of the work by Barankov and Levitov.
We agree that the detachment of the Higgs mode from the quasiparticle continuum has been observed by these colleagues previously in their preprint. Consequently, we added a brief discussion of their results as well.
We would like to highlight, that we propose additional modes beyond the finding of Barankov and Levitov, namely the existence of Higgs (and phase) modes beyond the primary ones for stronger interactions. We think that this represents an important novel finding.

We would like thank the Referee again for the interesting reference, thereby improving our manuscript by connecting it to previous research.

---

## Round 1 · Referee Report · Anonymous (Referee 2) · 2025-6-16

Strengths

  1. The paper considers an interesting and important subject: collective modes of strongly-coupled superconductors with realistic momentum-dependent interactions.

  2. Results are sound and are qualitatively reasonable, to the best of my judgement.

Weaknesses

  1. There is very little on the way of interpretation of the obtained results. For example, are the extra Higgs modes discovered by the authors correspond to radial (zero angular momentum?) bound-states of the two-particle problem with the same momentum-dependent pairing potential? Such two-particle problem can be solved analytically or numerically and results compared.

  2. The very similar situation was discovered 60+ years ago by Bardasis and Schrieffer. The authors mentioned them in one sentence. This is certainly not enough. They should articulate similarities and differences between their findings and those of BS much more explicitly.

  3. I find the use of static screening interactions (3) somewhat puzzling. The conventional treatment, known as "Tolmachev logarithm", calls for renormalizing the Coulomb line with the Cooper ladder, leading to: V_C --> V_C/(1+V_C\rho\log(E_F/\omega_D)) \approx 1/\rho*\log(...)
    The authors should explain what makes them deviate from this paradigm (usually thought to explain why phonon effect can ever overcome the Coulomb one).

Report

I am in favor of publishing the paper after the authors work on the above remarks.

Recommendation

Publish (meets expectations and criteria for this Journal)

  • validity: good
  • significance: good
  • originality: good
  • clarity: high
  • formatting: excellent
  • grammar: excellent

Author:  Joshua Althüser  on 2025-06-23  [id 5589]

(in reply to Report 2 on 2025-06-16)
Category:
answer to question
reply to objection

We thank the Referee for their time and kind and positive report as well as for the constructive suggestions. Let us address the points individually:

  1. Our calculations are done at zero angular momentum and at zero center-of-mass momentum. We have added a few explanatory sentences to make this clearer. We do not believe that there is an analytical solution to this problem because the effective interaction is not a density-density interaction in real space. This assessment is supported by the result from the reference (Barankov and Levitov [arXiv: 0704.1292]) provided by the other Referee, in which also a detachment of the Higgs mode from the lower edge of the two-particle continuum has been observed. This finding was obtained by numerical means as well.

  2. We agree that the results of Bardasis and Schrieffer deserve more discussion and have added a paragraph accordingly. The important distinction between the results by Bardasis and Schrieffer and ours in the manuscript consists in the angular momentum: We performed calculations at zero angular momentum L=0, while the Bardasis-Schrieffer modes hinge on L \neq 0.

  3. The usage of the screening (3) is a standard procedure known as the Thomas-Fermi screening. This type of expression arises from chains of bubble diagrams (cf. Fig. 1). In fact, this approach has been used before in the context of superconductors (cf. Simonato, Katsnelson, and Rösner in https://doi.org/10.1103/PhysRevB.108.064513). We further note that for a generic screening strength, we reproduce the results by Tolmachev, and Morel and Anderson. Hence, the renormalization of the interaction by the Cooper ladder. We added a paragraph below Eq. (3) discussing these connections.

We would like thank the Referee again for their aid in improving our manuscript by encouraging us to provide clearer explanations of our assumptions and calculations.

---

## Round 2 · List of Changes

Below Eq. (3): We added a brief discussion of the specific form of the screened Coulomb potential used and how it is related to the results by
by Tolmachev, and Morel and Anderson.

Below Eq. (23): We added a sentence, noting that the operators, and hence, the collective modes are isotropic and at zero center-of-mass momentum.

The paragraph just before section 4.3: We added that the detachment of the Higgs mode from the quasiparticle continuum has also been observed by Barankov and Levitov.

The bottom of page 21: We added a recap of the aforementioned results by Barankov and Levitov and compare our results to theirs explicitly.

The top of page 22: We added a paragraph elucidating the similarities and differences between the results obtained by Bardasis and Schrieffer and our results.

We added the following references:
[49] V. V. Tolmachev, Logarithmic criterion for superconductivity, Dokl. Akad. Nauk SSSR 140, 563 (1961).
[58] R. A. Barankov and L. S. Levitov, Excitation of the dissipationless Higgs mode in a fermionic condensate (arXiv:0704.1292) (2007), doi:10.48550/arXiv.0704.1292, 0704.1292.
[64] M. A. Müller and I. M. Eremin, Signatures of Bardasis-Schrieffer mode excitation in third-harmonic generated currents, Physical Review B 104(14), 144508 (2021), doi:10.1103/PhysRevB.104.144508.

---

## Editorial Decision

published